# Development and validation of the Self-Harm Screening Inventory (SHSI) for adolescents

Sojung Kim[1], Dong Gi Seo[2], Jae-Chul Park[3], Yeonkyeong Son[1], Ji-Hyun Lee[2], Dasol Yoon[3], Jae-Won Kim[4], Jae Hyun Yoo[5], Jong-Sun Lee[3] *

1 Department of Psychiatry, Hanyang University Seoul Hospital, Seoul, South Korea, 2 Department of Psychology, Hallym University, Chuncheon, Kangwon-do, South Korea, 3 Department of Psychology, Kangwon National University, Chuncheon, Kangwon-do, South Korea, 4 Division of Child and Adolescent Psychiatry, Department of Psychiatry & Behavioral Sciences, Seoul National University College of Medicine, Seoul National University Hospital, Seoul National University Children's Hospital, Seoul, South Korea, 5 Department of Psychiatry, The Catholic University of Korea Seoul St. Mary's Hospital, Seoul, South Korea

* jongsunlee@kangwon.ac.kr

**Data Availability Statement:** Data are available in the Supporting Information files.

**Funding:** This work was supported by the Ministry of Health & Welfare, Republic of Korea (grant number: HL19C0035).

## Abstract

Despite the rapidly increasing rate of non-suicidal self-injury (NSSI) among adolescents, there is a dearth of culturally appropriate psychological measures screening for NSSI among the adolescents in the Asian countries. This study aimed to develop and validate the Self-Harm Screening Inventory (SHSI), a culturally sensitive and suitable scale for screening adolescents for NSSI. In total, 514 Korean adolescents (aged 12–16 years) were recruited nationwide. All participants gave informed consent and completed the online self-report measures on NSSI, depression, anxiety, and self-esteem. Thereafter, preliminary items were developed through a series of steps: literature review, ratings of experts on self-harm and suicide, and statistical analyses. Ten of the 20 preliminary items were eliminated after exploratory factor analysis due to low endorsement and factor loading (less than .70). The final version of the SHSI comprised 10 binary items relating to self-harm behaviors within the past year (e.g., cut my body with sharp objects, hit my body). A confirmatory factor analysis supported a one-factor structure, as hypothesized. The one-factor model had a good model fit ($x^2(35) = 84.958$, $p < .001$, RMSEA = .053, CFI = .981, TLI = .975, SRMR = .124). The SHSI also had good internal consistency (Cronbach's alpha = .795) and 4-week test-retest reliability ($r = .786$, $p < .01$). The SHSI had high correlations with another self-harm related scale, the Self-Harm Inventory ($r = .773$, $p < .01$), and moderate correlations with the Child Depression Inventory ($r = .484$, $p < .01$) and Revised Children's Manifest Anxiety Scale ($r = .433$, $p < .01$). Additionally, the SHSI was negatively correlated with the Rosenberg Self-Esteem Scale ($r = -.399$, $p < .01$). The findings indicate that the SHSI is a reliable and valid measure for the screening of self-harm behaviors among adolescents.

## Introduction

Non-suicidal self-injury (NSSI) refers to "the deliberate, direct, socially unacceptable destruction or alteration of body tissue that occurs in the absence of suicidal intent" [1]. NSSI

**Competing interests:** The authors have declared that no competing interests exist.

behavior (for example, cutting, scratching, hitting or burning oneself) is relatively frequent among adolescents and young adults [2]. The onset of this behavior occurs between 12 and 14 years of age [3], and the behavior increases in prevalence during adolescence [4]. Research has found that around 10–23% of the adolescents in the general population, report the NSSI behaviors [3, 5–7]. However, the prevalence of NSSI among hospitalized adolescents in the clinical population is approximately 30%-50% [6, 8]. Girls are known to show NSSI earlier than boys and are at a higher risk of developing self-harm behaviors during adolescence [5, 9, 10]. NSSI declines over the course of its development from adolescence to early adulthood. However, approximately 20% of adolescents maintain the behavior for more than five years, and it often develops into a chronic and malignant practice that lasts until adulthood [4]. Individuals who engage in NSSI frequently and employ multiple NSSI methods to inflict greater physical damage, tend to show higher levels of psychological impairment, including various psychiatric disorders, risk-inducing behaviors, and suicidality [11, 12].

Researchers have found strong associations between NSSI and mental health problems, including depression, anxiety, and borderline personality disorder [13]. Although it starts as deliberate self-harm without the intention to die, repetitive NSSI could lead to suicide attempts over time. Individuals engaging in NSSI are known to cope with negative affect and stressful social situations using NSSI behavior. In addition, they experience negative reinforcement (e.g., decreased anger or sadness) and become desensitized to the pain associated with self-harming, which is a pattern related to suicidality [14, 15]. Consequently, the early detection of NSSI in adolescents is cardinal for its prevention and treatment.

The prevalence of NSSI among adolescents in Asia is rapidly increasing. In South Korea; 7.9% of middle school students and 6.4% of high school students reported a history of NSSI behavior in a recent nation-wide complete enumeration survey [16]. Empirical studies in Korea have shown that approximately 12.4–20% of middle school students reported self-injurious behavior [17, 18]. Additionally, the cases of adolescents seeking therapy due to self-harm has tripled over the last few years [19]. In Taiwan, NSSI ideation and behavior were observed in 15.1% and 9.7% of the adolescents aged 12–18 years, respectively [20]. Yet, research on NSSI among Asian adolescents is still in its infancy.

Research has shown that NSSI behavior among Asian adolescents is similar to that of the adolescents in the West, which often includes cutting, scratching, self-hitting, or banging the head against the wall [3, 13]. According to Lee [18], Korean middle school students reported engaging in self-scratching (78.35%), wrist-cutting (67%), and hitting their head against the wall (23.7%), as the top three forms of self-injurious behaviors. In a study on self-harm among female middle school students [17], hitting oneself was the most common method (17.1%), whereas leaving wounds untreated (15.4%), scratching one's body with a sharp object (13.7%), and hitting one's head against a wall or a desk (13.1%) were other commonly endorsed methods. However, a few differences exist. Unlike Western adolescents, some Korean adolescents use unique methods for NSSI, such as "saheol," or stitching one's hands. Saheol translates to bloodletting and refers to a practice wherein Korean adolescents buy injection needles to puncture their skin and collect blood or bloody tissue paper. This practice became popular since it is considered to carry a low risk of getting caught as it leaves few scars. Moreover, it is considered hygienic as sterilized needles are used and is an effective strategy to impress peers by uploading pictures of it on social networking sites and online blogs. Some reports have shown that blood-letting is one of the most common NSSI methods among adolescents who harm themselves [21, 22]. Thus, it seems necessary to include culturally sensitive items in the screening measures to detect the NSSI behavior in adolescents from diverse cultures and regions.

For the last two decades, a wide variety of instruments have been developed to assess self-harm behaviors. However, there are only a few validated and peer-reviewed measures of self-

injury. One of the valid self-report measures is the Self-Harm Inventory (SHI) [23]; it is the first and widely used measure for assessing self-harm behaviors. It consists of 22 binary items and detects borderline personality disorder at a hit rate of 87.9%. The Deliberate Self-Harm Inventory (DSHI) is another popular measure of NSSI [24]. It consists of 17 items focused on the explicit behavioral aspects of NSSI, asking about the method, frequency, and period of NSSI behaviors in the respondent's lifetime. While the SHI and DSHI focus on NSSI behaviors, several other measures assess the multidimensional aspects of NSSI. For example, the Functional Assessment of Self-Mutilation (FASM) asks about the method, frequency, and treatment for NSSI, as well as the motivations for self-harm [25]. Furthermore, the Inventory of Statement About Self-Injury (ISAS) evaluates the self-injurious behaviors, their function, and the respondent's statements on self-injury [26]. These self-report measures are high on psychometric soundness [27]; however, they are not always the best choice for assessing self-harm behaviors among adolescents, for several reasons.

First, most of the aforementioned inventories have been validated using young adult samples, and some items (such as those in the SHI on "reckless driving" and "unsafe sexual relationships") are not appropriate for use with adolescents in the Asian countries. Furthermore, all the three measures were validated in Korea (FASM, K-ISAS, and K-SHI) based on the data from young adults [28–30]. However, currently, there is no valid measure for assessing self-harm behaviors in Korean adolescents. Second, although the aforementioned measures focus on NSSI behaviors, some of them do not directly reflect observable behaviors. For example, some items of the SHI (items 11, 12, 14, 15, 17, and 20) explore interpersonal, occupational, and religious aspects of these behaviors, rather than explicitly assessing self-harm behaviors. In addition, measures—such as the FASM and ISAS—that able to obtain comprehensive information on self-harm behaviors, including the motivation, functions, and effects of self-injury, could be limited in their utility as quick and efficient screening tools. Finally, the existing measures have primarily been developed in the Western countries, and culturally sensitive items have been seldom developed or validated for use in the Asian countries. In Korea, researchers have translated and validated several measures of self-harm [28–30]. However, no Korean measure has been developed to assess the self-harm behaviors in the Korean population. Likewise, the Japanese version of the DSHI had been validated [31]. However, originally developed Japanese measures to assess self-harm behaviors in Japan, are hard to seek. Although Chinese research has reported a few self-harm measures developed in China, the validation of the measures is rare [32–34]. Considering the potential gap between self-harm behaviors in the Asian and Western cultures, and the lack of reliable tools for assessing specific self-harm behaviors in Asia, it seems necessary to develop items reflecting self-harm behaviors in the Eastern countries.

This study aimed to develop a brief screening measure for self-harming behaviors among adolescents. Based on the literature review and clinical observation, we devised preliminary items and developed a screening measure for self-harm behaviors to detect the NSSI behaviors at their age of onset. The factor structure and psychometric properties of the tool were investigated among community-dwelling adolescents in Korea.

## Methods

### Participants

We recruited participants on a nationwide scale from multiple elementary and middle schools in South Korea. Participants were either introduced to this study during their classes or were provided with the information on the bulletin board at school. A total of 514 students—11 students in grade 6 from three elementary schools, and 503 students in grades 1–3 from six middle

schools in Seoul, Gyeonggi, Kangwon, and Geolla areas—participated in our study. Grade 1–3 in middle school in South Korea is comparable to the grade 7–9 in middle school in Western countries. The participants' mean age was 13.71 ($SD$ = 0.911), and their age range was 11–16 years. Among the participants, 269 (52.3%) identified as female, and 228 (44.4%) identified as males, while the other 17 participants (3.3%) did not provide information on their gender.

## Measures

**The Self-Harm Screening Inventory (SHSI).**   The Self-Harm Screening Inventory (SHSI) is a brief self-report measure for assessing self-harm behaviors among adolescents. The SHSI consists of 10 binary items (yes/no) which inquire about one's engagement in self-harm behaviors within the past year. For the purpose of assessing NSSI among adolescents, the definition of NSSI was provided at the end of the instructions as, "Non-suicidal self-injury refers to deliberate self-harm behavior without the intention to die." The participants were instructed to choose "yes" if they had engaged in self-harm at least once in their lifetime, and to choose "no" if they had not. The Cronbach's alpha of the SHSI for this sample was .795. The final version of the SHSI is presented in S1 Table.

**The Self-Harm Inventory (SHI).**   The SHI was developed by Sansone, Wiederman, and Sansone [23] to measure the intentional self-harm. It consists of 22 binary items (yes/no) on suicide attempts, overdose, cutting oneself on purpose, burning oneself on purpose, set oneself up in a relationship to be rejected, etc. The current study adopted the Korean version of the SHI (K-SHI), which was validated by Kim, Woo, Koo, and Lee [29]. They reported sound internal consistency of the items with a Cronbach's alpha of .76 [29]. Eight items, with an item-total correlation lower than .300, were excluded from this study. The Cronbach's alpha for the remaining 14 items was .780.

**The Children's Depression Inventory (CDI).**   The Children's Depression Inventory (CDI) was developed by Kovacs [35] to assess the severity of depressive symptoms in children and adolescents aged 7–17 years. It consists of 27 items that are assessed using a 3-point Likert scale (ranging from 0–2). In their validation of the Korean version of the CDI, Cho and Lee [36] reported good internal consistency with a Cronbach's alpha = .88. In this study, four items were excluded because of low item-total correlations ($r < .300$). The Cronbach's alpha for the remaining 23 items was .887.

**The Revised Children's Manifest Anxiety Scale (RCMAS).**   Reynolds and Richmond [37] developed the Revised Children's Manifest Anxiety Scale (RCMAS) to measure the symptoms of anxiety among children and adolescents. The RCMAS consists of 37 binary items (yes/no). A high total score reflects high levels of anxiety. Choi and Cho [38] validated the RCMAS in Korea, with a Cronbach's alpha of .81. Ten items were excluded from the current study due to low item-total correlations and relevance of their contents (e.g., fictitious items). The Cronbach's alpha for the final 27 items was .902.

**The Rosenberg Self-Esteem Scale (RSES).**   The Rosenberg Self-Esteem Scale (RSES) is the most widely used measure to assess self-esteem among adults, adolescents, and children. It was developed by Rosenberg [39] and consists of 10 items. The internal consistency of the Korean version of the RSES, translated by Jeon [40], was found to be acceptable in a study conducted by Lee [41]. In the current study, one item with low item-total correlation was excluded from the analysis. The Cronbach's alpha for the other nine items was .994.

## Procedures

For the development of the preliminary SHSI, multiple sources of information were reviewed. Previous research findings (academic sources), news, and social media (such as Twitter,

Instagram, Facebook, and Tumblr), describing self-harm behaviors and related clinical observations, were investigated. The comments of experts on self-harm and suicide attempts (non-academic sources) were also considered. Further, the items assessing self-harm behaviors in the pre-existing self-harm measures, such as the SHI, DSHI, and FASM, were collected. The focus of the literature review was to generate a list of specific and observable NSSI behaviors. Thereafter, the overlapping items and dimensions in the list were either combined or reorganized.

A total of 28 items were developed for the first version of the preliminary measure (S2 Table) and sent to 11 Korean experts—four clinical psychologists, five psychiatrists, and two school counselors—in the areas of self-harm and suicide. Furthermore, to ease comprehension among the readers, we have added the preliminary 28 items under the Supplemental Materials. The experts rated individual items for their *appropriateness* (in assessing self-harm among adolescents), *sensitivity* (to distinguish the adolescents engaging in serious self-harm), and *prevalence* (of the behavior among Korean adolescents). They rated each item on a seven-point Likert scale (1 = very unlikely; 7 = very likely) for *appropriateness* and *sensitivity*, and a three-point Likert scale (1 = low prevalence; 3 = high prevalence) for *prevalence*. They also provided additional comments on the utility of the items. Thereafter, the mean rating scores for each item were calculated. First, items with scores lower than 5 for *appropriateness* and *sensitivity*, and scores lower than 2 for *prevalence*, were identified. Of the 28 items, eight items below the aforementioned scores were identified under all three dimensions, one item under two of the dimensions, and six items under only one dimension. Items identified in more than two of the dimensions were excluded. Second, the informative value of the excluded items was reconsidered in the ensuing discussion among the authors. Of the nine items identified under more than two dimensions, the item on "overdose" was considered a distinctive and comparatively common method of self-harm. It was, thus, re-included among the preliminary items. Consequently, eight items with low scores on more than two of the dimensions (appropriateness, sensitivity, and prevalence) were discarded before administration (e.g., items on binge drinking, having unsafe/multiple sexual relationships, starving, laxatives/diuretics abuse, rubbing skin on bumpy objects, swallowing a toxic substance, or getting a tattoo). Twenty items with higher scores and utility were retained as preliminary items for the SHSI and administered to the participants along with other self-report measures.

All the participants voluntarily participated in the study. Participants and their parents were informed about the purpose of the study and its procedures. The research assistants provided additional information about the study on the request of the participants or their parents. Written informed consent was obtained from the participants and their parents prior to participation. A URL of the online survey webpage was sent to the participants' smartphones. Participants logged into a secure webpage and completed a set of self-report measures containing the preliminary versions of the SHSI, K-SHI, CDI, RCMAS, and RSES. It took them approximately 20 minutes to complete all the questionnaires, and they received a monetary reward after submitting the measures. To estimate the temporal stability of the preliminary SHSI, the same tool was administered to 97 participants, four weeks after the first administration. Participants received a gift card worth 5,000 won (approximately 4 dollars) for their participation in the first administration and an additional 3,000 won (approximately 3 dollars) worth gift card after the second administration. All research procedures were approved by the Institutional Review Board of the Kangwon National University in Chuncheon, South Korea (IRB NO. KWNUIRB-2019-05-007-002). The research procedures have been presented in Fig 1.

## Statistical analysis

The primary purpose of this study was to develop a scale for screening self-harm behavior among Korean adolescents. Item and factor analyses were performed on the preliminary 20

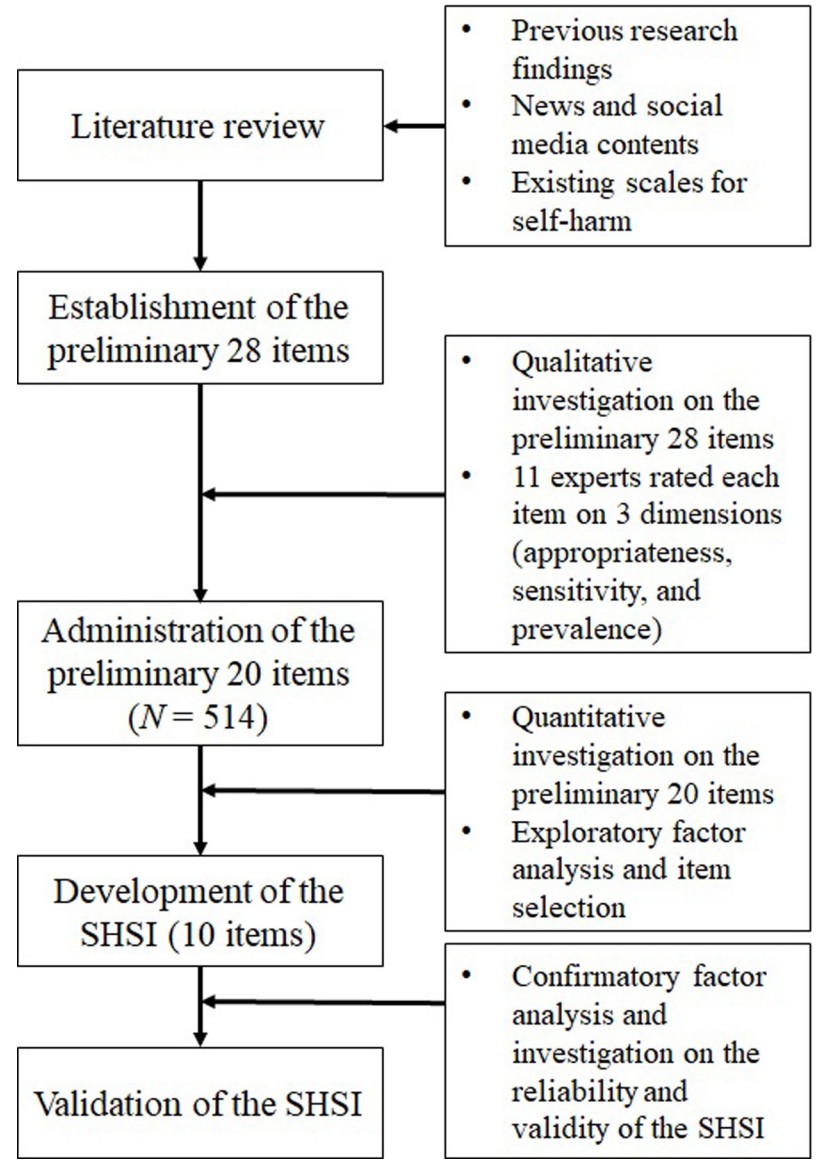

**Fig 1. Flow of the research.**

items to select appropriate items for the SHSI. Thereafter, preliminary analyses were conducted and the psychometric properties of the items were investigated to develop the final version of the SHSI. Preliminary analyses refer to the process of correcting and deleting items based on their psychometric properties, aimed at developing a scale with reliable and valid items related to its purpose. In the item analyses, not only items of the developing scale, but also items from the criterion scales should be tested [42]. In this study, preliminary item analyses for 20 items of the SHSI were conducted, and five criterion scales were used to select valid items using a representative sample.

The item-total correlations and internal consistency coefficients were determined using the IBM SPSS 25 software. If an item-total correlation value was found to be lower than 0.3, or the internal consistency coefficient was extremely low (marked by factor loading lower than 0.7)

[43, 44], then the item was removed from the initial screening scale for self-harm behavior. The items were finalized after three iterations of item analyses.

After the item analyses, factor analyses were conducted to examine the factor structure of the SHSI, and to determine the items to be included in the final scale. Both exploratory factor analysis (EFA) and confirmatory factor analysis (CFA) were implemented using Mplus 8.0 [45, 46]. Assuming that the factors were interrelated, GEOMIN rotation was conducted to find the best-fitting model for the EFA. As the SHSI comprised binary items and assessed self-harm behavior, which is sporadically observed in a community, the WLSMV (weighted least squares with means and variances adjusted) estimation method was used. The WLSMV method is appropriate for use when non-normal distributions are expected for the observed variables [45]. The factor structure for the preliminary SHSI was determined based on the model fit indices and interpretability [47]. Tucker Lewis Index (TLI), Comparative Fit index (CFI), Root Mean Square Error of Approximation (RMSEA), and Standardized Root Mean Square Residual (SRMR) were applied for the alternative models.

Final items for the SHSI were determined based on the result of the EFA. First, items with factor loading above 0.7 were considered for inclusion in the final SHSI. Then, each item was evaluated for interpretability and confirmed by two clinical psychologists. CFA was conducted to validate the final version of the SHSI.

Descriptive analyses were conducted to investigate the frequencies of our sample endorsing the preliminary and final versions of the SHSI. Gender differences in the endorsement rate and mean scores of the SHSI items were tested via the $X^2$ test and student's $t$ test. Pearson's $r$ was used for test-retest reliability and item-total correlation. Cronbach's alpha was determined to verify the internal consistency of the SHSI. In all the statistical analyses, two-tailed tests were used, and the significance of the calculated coefficients was defined as $p < .05$.

## Results

### Development of the SHSI

To test the appropriateness of the data from our sample, the Square Multiple Correlation (SMC) and polychoric correlation coefficients of the preliminary items were analyzed. The results showed that there were no significant problems of correlation coefficients or multicollinearity (SMC > .800) in our data.

The endorsement rate for the items ranged between 0.6% and 28.2%. The preliminary item 12, "Bit parts of my body (e.g., mouth and lips.)," was the most frequently endorsed item. The preliminary items 10, "Burnt skin with fire (using cigarette, match, or other hot objects)," and 16, "Cut holes in my body (e.g. not ear piercings or body piercings that are meant to enhance beauty, but to make holes to harm myself), were both endorsed by just one participant from the sample. Three items (preliminary items 10, 11, and 16) had low endorsement and low item-total correlations ($r < .30$, $p < .05$) and were excluded from further analyses. Refer to Table 1 for the frequencies of our sample endorsing the preliminary 20 items of the SHSI.

Before conducting the EFA, the Kaiser-Meyer-Olkin (KMO) test and Bartlett's test of sphericity were conducted on the remaining 17 items to investigate the appropriateness of the data for the factor analyses. The KMO measure of .832, which is greater than .6, signifies that the sample from which these data were collected, was adequate [48]. Bartlett's test of sphericity yielded statistically significant results, at $p < 0.01$. The SMCs for all the variables ranged between .018 and .747, which is below .800. These results indicated that our sample was adequate for further analyses [49].

**Table 1. Frequencies of the total sample (N = 514) endorsing the preliminary SHSI.**

| | SHSI preliminary items | Total N (%) | |
|---|---|---|---|
| | | Yes | No |
| 1 | Overdosed on drugs. * | 6 (1.2) | 508 (98.8) |
| 2 | Cut my body with sharp objects. * | 28 (5.4) | 486 (94.6) |
| 3 | Hit my body (e.g., hitting my body, such as the head hard, with my hands). * | 63 (12.3) | 451 (87.7) |
| 4 | Banged my head against a wall, desk, etc. * | 39 (7.6) | 475 (92.4) |
| 5 | Hit things hard with my fist. | 111 (21.6) | 403 (78.4) |
| 6 | Scratched my body. * | 22 (4.3) | 492 (95.7) |
| 7 | Cut or carved something onto skin using a knife. * | 20 (3.9) | 494 (96.1) |
| 8 | Pulled my hair out. | 59 (11.5) | 455 (88.5) |
| 9 | Picked or pinched my wound. | 17 (3.3) | 497 (96.7) |
| 10 | Burnt my skin with fire (using a cigarette, match, or other hot objects). | 1 (0.2) | 513 (99.8) |
| 11 | Stuck objects underneath my fingernails or into my skin. | 7 (1.4) | 507 (98.6) |
| 12 | Bit parts of my body (e.g., mouth and lips) | 145 (28.2) | 369 (71.8) |
| 13 | Scratched my skin until it left scars. | 32 (6.2) | 481 (93.6) |
| 14 | Picked or peeled off my skin. | 59 (11.5) | 455 (88.5) |
| 15 | Stabbed my body with sharp or pointed objects. * | 4 (0.8) | 509 (99.0) |
| 16 | Cut holes in my body (e.g., not ear piercings or body piercings that are meant to enhance beauty, but to make holes to harm myself). | 1 (0.2) | 513 (99.8) |
| 17 | Slit or cut my body with sharp objects. * | 18 (3.5) | 493 (95.9) |
| 18 | Carved words or symbols onto my body. | 9 (1.8) | 505 (98.2) |
| 19 | Strangled my neck. * | 9 (1.8) | 504 (98.1) |
| 20 | Engaged in bloodletting (drained blood from my body). * | 3 (.6) | 510 (99.2) |

* Items included in the final version of the SHSI.

The EFA was conducted on the remaining 17 items to investigate the factor structure of the preliminary SHSI. The number of factors was determined by the Kaiser's rule [49], the scree plot, and the ratio of the explanatory variances.

On conducting the EFA with GEOMIN rotation, three components showed eigenvalues higher than 1 and explained 76% of the total variance. Consequently, three models (1-, 2-, and 3-factor models) were selected as hypothesized models. The model fit indices for each model have been presented in Table 2.

Although all the models showed good fit, the 1-factor model with 17 items was determined as the final model for the SHSI, based on model interpretability.

The final structure showed a good model fit: $\chi^2$ (119) = 207.189, $p$ < .001, RMSEA = .038, CFI = .964, TLI = .959, WRMR = 1.154. Factor loadings for the total items ranged between .555 and .985. Among the 17 items, 10 items (preliminary items 1, 2, 3, 4, 6, 7, 15, 17, 19, and 20), which showed high factor loadings (greater than 0.7), were examined by two authors for content validity, and were selected for the final version of the SHSI.

**Table 2. Results from the exploratory factor analyses of the preliminary SHSI (N = 514).**

| Model | $\chi^2$($df$) | $p$ | RMSEA | CFI | TLI | SRMR |
|---|---|---|---|---|---|---|
| 1-factor model | 84.958(35) | .0000 | .053 | .981 | .975 | .124 |
| 2-factor model | 25.468(26) | .493 | 0 | 1 | 1 | .056 |
| 3-factor model | 10.600(18) | .911 | 0 | 1 | 1.007 | .027 |

## Prevalence of self-harm behavior

Approximately 19% of the participants ($N$ = 95) reported engaging in self-harm in the last year (SHSI total score > 0). Among them, 8.2% reported using one method, 6% reported adopting two methods, and 4.3% ($N$ = 22) reported adopting more than three methods of self-harm, in the last year. To assess the lifetime history of self-harm behaviors, the total score of the SHI was used. Among the 514 participants, 22% reported engaging in self-harm in their lifetime (total score > 0). Approximately 12% of the participants reported having used multiple methods, and a third used four or more methods to harm themselves (Table 3).

To examine the difference between genders, the means of the total SHSI scores were compared. Girls showed significantly higher mean scores than boys, $t(438.404)$ = -2.094, $p$ = 0.037. To compare the prevalence of self-harm between genders, cross analysis was administered. Participants were coded as "Non-self-harm group" when their total SHSI score was 0. Participants who scored 1 or higher on the SHSI, were coded as "Self-harm group". Results showed that girls were at an increased risk of being included in the self-harm group, $\chi^2$ (1) = 3.843, $p$ = 0.050.

Among the individual items, item 3, "Hit my body (e.g., hitting my body, such as the head, hard with my hands)," and item 4, "Banged my head against a wall, desk, etc." were the most frequently endorsed self-harm methods. Among the girls, item 6, "Cut or carved something onto skin using a knife," was the next most reported method, while the boys reported item 5, "Scratched my body" as their third common method. The between-group difference analyses indicated that the girls showed significantly higher endorsement for items 2, 6, and 8, than the boys (Table 4).

## Factor structure, reliability, and validity of the SHSI

To investigate the psychometric properties of the SHSI, the factor structure, reliability, and validity of the final version were analyzed. For the final 10 items of the SHSI, a CFA was conducted with GEOMIN rotation. The one-factor model also showed a good model fit for the 10 items: $\chi^2(35)$ = 84.957, $p$ < .001, RMSEA = .053, CFI = .981, TLI = .975, WRMR = 1.077. The factor loadings of the 10 items ranged between .777 and .992 (Table 5).

To investigate the internal consistency of the test items, the item-total correlations and Cronbach's alpha were analyzed (Table 6). The item-total correlations for each item were acceptable, ranging between .310 and .670. The Cronbach's alpha was good ($\alpha$ = .795).

**Table 3. Total SHSI score for the total sample, girls, and boys (N = 514).**

| Total SHSI Score | Total (N = 514)* N (%) | Girls (n = 269) n (%) | Boys (n = 228) n (%) |
|---|---|---|---|
| 0 | 419 (81.5) | 108 (77.3) | 195 (85.5) |
| 1 | 42 (8.2) | 28 (10.4) | 13 (5.7) |
| 2 | 31 (6.0) | 20 (7.4) | 11 (4.8) |
| 3 | 8 (1.6) | 2 (0.7) | 6 (2.6) |
| 4 | 3 (0.6) | 1 (0.4) | 2 (0.9) |
| 5 | 3 (0.6) | 3 (1.1) | 0 (0) |
| 6 | 3 (0.6) | 2 (0.7) | 1 (0.4) |
| 7 | 3 (0.6) | 3 (1.1) | 0 (0) |
| 8 | 3 (0.2) | 1 (0.4) | 0 (0) |
| 9 | 0 (0) | 0 (0) | 0 (0) |
| 10 | 1 (0.2) | 1 (0.4) | 0 (0) |
| Mean (SD) | 0.41 (1.162) | 0.54 (1.397) | 0.29 (0.838) |

*17 Participants who did not report their gender were included in the sample.

**Table 4. Frequencies of the total sample, girls, and boys endorsing the SHSI (N = 514).**

| Items | Total ($N$ = 514) $^†$ N (%) | Girls ($n$ = 269) $n$ (%) | Boys ($n$ = 228) $n$ (%) | $\chi^2$ |
|---|---|---|---|---|
| **1** | 6(1.2) | 4 (1.5) | 2 (0.9) | .385 |
| **2** | 28(5.4) | 22 (8.2) | 6 (2.6) | 7.142** |
| **3** | 63(12.3) | 40 (14.9) | 23 (10.1) | 2.550 |
| **4** | 39(7.6) | 21 (7.8) | 18 (7.9) | 0.001 |
| **5** | 22(4.3) | 15 (5.6) | 7 (3.1) | 1.832 |
| **6** | 20(3.9) | 17 (6.3) | 3 (1.3) | 8.000** |
| **7** | 4(0.8) | 4 (1.5) | 0 (0) | 3.403 |
| **8** | 18(3.5) | 14 (5.2) | 4 (1.8) | 4.098* |
| **9** | 9(1.8) | 5 (1.9) | 3 (1.3) | 0.235 |
| **10** | 3(0.6) | 2 (0.7) | 1 (0.4) | 0.188 |

$^†$17 Participants who did not report their gender were included in the sample.

** $p < .01$.

To assess the temporal stability of the inventory, 97 participants completed the SHSI four weeks after the first administration. There was no significant difference in the ages ($t$(92) = -.971, $p > .05$) or the gender ($\chi^2$(10) = 13.591, $p > .05$) between the total sample and the participants who completed the retest. The four-week test-retest reliability was good ($r$ = .786, $p < .01$).

The SHSI had high correlations with the SHI ($r$ = .775, $p < .01$), and moderate correlations with the CDI ($r$ = .489, $p < .01$) and RCMAS ($r$ = .433, $p < .01$). The SHSI was negatively correlated with the RSES ($r$ = -.399, $p < .01$; Table 7). The results indicate that SHSI is a reliable and valid measure for NSSI, and it is distinct from the other psychological measures assessing depression, anxiety, and self-esteem (Table 7).

## Discussion

The purpose of this study was to develop and validate a culturally sensitive screening measure for self-harm, reflecting the key self-harm behaviors among early adolescents. The

**Table 5. Results from the confirmatory factor analysis of the SHSI (N = 514).**

| No. | Items | Factor loadings |
|---|---|---|
| **1** | Overdosed on drugs. | .800* |
| **2** | Cut my body with sharp objects. | .972* |
| **3** | Hit my body (e.g., hitting my body, such as the head hard, with my hands). | .843* |
| **4** | Banged my head against a wall, desk, etc. | .832* |
| **5** | Scratched my body. | .859* |
| **6** | Cut or carved something onto my skin using a knife. | .928* |
| **7** | Stabbed my body with sharp or pointed objects. | .884* |
| **8** | Slit or cut my body with sharp objects. | .992* |
| **9** | Strangled my neck. | .777* |
| **10** | Engaged in bloodletting (drained blood from my body). | .862* |
| | Eigenvalue | 7.320 |
| | Total variance (%) explained | 72.930 |

* $p < .05$.

** $p < .01$.

**Table 6. Mean, standard deviation, and item-total correlations of the SHSI (N = 514).**

| No. | Items | M | SD | Item-total correlations |
|---|---|---|---|---|
| 1 | Overdosed on drugs. | 0.01 | 0.11 | .383 |
| 2 | Cut my body with sharp objects. | 0.06 | 0.23 | .622 |
| 3 | Hit my body (e.g., hitting my body, such as the head hard, with my hands). | 0.12 | 0.33 | .483 |
| 4 | Banged my head against a wall, desk, etc. | 0.08 | 0.27 | .454 |
| 5 | Scratched my body. | 0.04 | 0.20 | .607 |
| 6 | Cut or carved something onto my skin using a knife. | 0.04 | 0.19 | .609 |
| 7 | Stabbed my body with sharp or pointed objects. | 0.01 | 0.09 | .382 |
| 8 | Slit or cut my body with sharp objects. | 0.04 | 0.19 | .670 |
| 9 | Strangled my neck. | 0.02 | 0.13 | .424 |
| 10 | Engaged in bloodletting (drained blood from my body). | 0.01 | 0.08 | .310 |
| | Cronbach's alpha | | .795 | |

psychometric properties of the SHSI were investigated, and the reliability and validity of the measure were found to be good.

As our participants were recruited from multiple elementary and middle schools nation-wide, and the total sample had equal gender proportions, the data from this sample might well reflect the phenomena related to self-harm among the adolescents in South Korea. According to the total SHI scores in our sample of adolescents between 11 and 16 years of age, 22% of the young adolescents have engaged in self-harm behavior in their lifetime. Based on the total score of the final 10 items of the SHSI, 18.5% of the adolescents reported engaging in deliberate self-harm behavior in the last year; 22.7% of girls used at least one self-harm method, while 14.5% of the boys also engaged in these behaviors. More than 4% of the participants obtained a total score of 3 or higher on the SHSI, reflecting that they had engaged in multiple self-harm methods in their lifetime. The prevalence was similar but slightly higher than we expected in young adolescents in South Korea, since previous research have reported the prevalence of self-harm behavior for these groups to be between 7.9% and 20% [16–18]. These numbers are also comparable to the prevalence of self-harm among the adolescents in Western countries, which is around 10% to 23% [3, 5–8]. These findings indicate that, while the adolescents in Asia have received much less research attention, they have a high prevalence rate of self-harm behaviors, and a number of them also engage in multiple methods of self-harm, which is directly related to more serious forms of self-harm [11]. As early adolescence is a period when problematic behaviors evolve into chronic self-harm, it could contribute to the increase in the prevalence of non-suicidal self-harm, and ultimately lead to suicide attempts later in life.

**Table 7. Correlations with related measures.**

| | SHSI | SHI | CDI | RCMAS |
|---|---|---|---|---|
| **SHI** | .775** | - | | |
| **CDI** | .489** | .546** | - | |
| **RCMAS** | .433** | .479** | .717** | - |
| **RSES** | -.399** | -.449** | -.787** | -.602** |
| **M (SD)** | .41(1.16) | .53(1.39) | 8.34(6.74) | 8.29(6.12) |

SHSI = Self-Harm Screening Inventory; SHI = Self-Harm Inventory; CDI = Child Depression Inventory;
RCMAS = Revised Children's Manifest Anxiety Scale; RSES = Rosenberg Self-Esteem Scale.
** $p < .01$.

In terms of gender, our research participants showed significant differences in the prevalence of self-harm behavior not only with respect to the aforementioned frequencies of self-harm behavior, but also pertaining to the mean SHSI scores, which were found to be significantly higher among the girls. Additionally, the incidence of any form of self-harm behavior was marginally higher among adolescent girls than among boys, $x^2(1) = 3.843$, $p = 0.050$. These results indicate that similar to the Western countries, Asian girls are at a higher risk of engaging in self-harm behavior. Most research findings showed female adolescents and young adult to be 1.5 to 3.0 times more likely to deliberately injure themselves, than their male peers [50, 51]. In the same vein, girls showed higher endorsement of almost every item in the SHSI. There were significant gender differences at the individual item level as well. Among the final 10 items of the SHSI, three items (2. Cut my body with sharp objects, 6. Cut or carved something onto my skin using a knife, and 8. Slit or cut my body with sharp objects) were found to have significant gender differences in their endorsement. The three items, which were endorsed more often by the girls, had in common the practice of self-cutting, which is consistent with the findings from previous research, stating that girls prefer to use self-harm methods related to cutting, scratching, and the sight of blood [51, 52]. Only one method was slightly higher in its endorsement among the boys, namely, item 4, "Banged my head against a wall, desk, etc." This was also consistent with previous findings that boys prefer to hit themselves [51, 53]. While the most frequently used self-harm methods (such as hitting and banging their head) among both boys and girls were less severe in terms of the risk involved, cutting and carving on their bodies with a knife was the third most frequent method of self-harm among the girls, as opposed to the scratching method reported by the boys. Thus, not only do the girls seemed to endorse self-harm behaviors more frequently, but they also used more lethal ways to hurt themselves, as compared to the boys. However, further investigation of the observed gender gap is beyond the scope of the current research.

Whitlock et al. [51] found similar gender differences in the frequencies and forms of NSSI behaviors in the college-going population, and also noticed differences in the function and initial motivations for NSSI. Female college-going students were significantly more likely than the males to experience an overwhelming need to use NSSI as a form of self-control to regulate their affective states. Women also reported being upset or hoping that someone would notice their self-injury [51]. In consideration of the findings of Andover, Pepper, and Gibb [9], it is evident that the individuals with an NSSI history tend to utilize *avoidant* coping strategies significantly more often than their peers who do not engage in self-harm. Moreover, the females who engaged in self-harm endorsed the use of problem solving and social support-seeking strategies less often than the females without any NSSI history. These findings suggest that the strategies for emotion regulation and coping should be more intentionally delivered to the girls in treatment for NSSI [51].

While developing the preliminary items for the SHSI, the inclusion of frequently observable and culturally sensitive self-harm behaviors was of central importance. To achieve this goal, a list of items was carefully devised, following review. The existing measures of self-harm, such as the SHI, DSHI, and FASM, are based on years of clinical observations, the testimonies of individuals who engaged in self-harming behavior, and the behaviors reported in literature [23, 24]. They consist of excellent items for reflecting self-harm behaviors in real life, with high reliability and validity. However, these measures do not clearly highlight the process of selecting items in their development and validation studies [24]; item selection for these measures is wholly based on the correlation of the test items with a related psychological construct such as borderline personality disorder [23]. These approaches might lack objectivity in selecting items and, therefore, we conducted a series of qualitative and statistical analyses to select the final items for the SHSI.

First, we developed items based on the review of academic and non-academic sources on core areas of self-harm among adolescents. We also considered the ratings of experts on NSSI and suicide for objective assessment of the preliminary items. Items were rated on three dimensions: appropriateness, sensitivity, and prevalence. Among the preliminary 28 items, 20 items with high ratings in the quantitative assessment, and importance as suggested by the qualitative assessments, were selected to be administered to the participants. After the administration, three items with low endorsement and low item-total correlations were excluded. Exploratory factor analyses were carried out for the remaining 17 items; the 1-factor model was adopted based on model fit indices and the theoretical assumption that our measure assesses explicit self-harm behaviors, with reference to measures such as the SHI [23] and DSHI [24]. To devise a robust and brief screening measure, we selected 10 items with factor loadings above .7. Finally, the 1-factor model was also found to fit the final 10 items well. With the item development and selection process, we examined the psychometric properties of each item and screened informative items both in terms of statistics and content. The SHSI has its strength in parsimonious and effective adaptation for adolescents engaging in self-harm. Furthermore, the final version of the SHSI was found to have good reliability, validity, internal consistency, and temporal stability. It also showed a high correlation with the SHI, moderate correlations with the measures of depression and anxiety, and a negative correlation with the measure of self-esteem.

At the individual-item level, a few preliminary items showed high endorsement but were excluded from the final SHSI because of their low factor loadings, namely, preliminary items 12, "Bit parts of my body (e.g., mouth, lips, etc.)" (28.2%), 5, "Hit things hard with my fist" (21.6%), 8, "Pulled my hair out" (11.5%), and 14, "Picked or peeled off my skin" (11.5%). However, these self-harming behaviors were consistently found in previous Korean research. For example, biting body parts was the most common method for self-harm in two college populations [28, 30] and one high school sample [54]. Approximately 26.5% to 62.1% of the college student, and 11% of the high school students reported engaging in biting as a self-harm method. Hair-pulling was another frequently reported method of self-harm, endorsed by 24.5% of the college sample [28] and 8% of the high school sample [54]. While hitting things hard with a fist, and picking or peeling skin off, were not as popular in previous studies, all the four aforementioned methods do not have consequences as severe as the other methods included in the final SHSI. Considering that the previous findings suggest that serious self-harm methods are used among adolescents in clinical setting [55], the SHSI—which considered factor loading as the criteria and not the frequency—might have decreased the false-positive errors in selecting the adolescents with self-harming behaviors.

A few items—1. Overdosed on drugs (1.2%); 7. Stabbed my body with sharp or pointed objects; and 10. Engaged in bloodletting (drained blood from my body) (0.6%)—were included in the final version of the SHSI, in spite of low endorsement by the sample. These items showed high factor loadings (above .80). Considering the level of lethality of these items, they might be able to screen individuals engaging in more severe methods of self-harm. Drug overdose is particularly known to be related to repetitive suicidal attempts. Research on suicide has demonstrated that individuals who repetitively attempt suicide, tend to have chronic psychiatric disorders and engage in overdose as their preferred method (53). Adolescents who overdose themselves on drugs at their onset age for NSSI, might be at a higher risk of attempting suicide than their peers using less serious methods to harm themselves. Additionally, bloodletting, which was included as a culturally sensitive item, showed high factor loading. To the best of our knowledge, bloodletting is a unique way of self-harming and has never been included in any other self-harm related measures. Although, the method was primarily reported in the non-academic sources in Korea [21, 22], it might be worth investigating whether bloodletting is used as a self-harm method in other Asian or Western countries.

Based on the findings of this study, we believe that future studies need to examine the endorsement rate of the SHSI in a clinical population. In this study, we developed and validated the SHSI based on the data from early adolescents in the non-clinical population. Thus, a strength of this measure is that it has been empirically validated, and thus could be easily adopted for use with communities. Future research may benefit from the utilization of the SHSI with clinical adolescents and adults. It may also benefit from the observation of the differences in the endorsement rate of each item, and the relationships between the SHSI and other thought, urges, or behaviors of self-harm (e.g., methods of implicit or indirect self-injury).

However, there are a few limitations to be noted. First, the details of specific self-harming behaviors and related cognitive and emotional variables were not intensively collected from the participants. The authors reviewed the literature and employed expert ratings to determine the valid items. Since the study employed the survey method of data collection, direct self-harm reports from the adolescents could not be availed. Therefore, future research should employ different research methods, such as semi-structured or focus group interviews, to investigate specific self-harm behaviors and the related cognitive and emotional variables (e.g., motivation and functions of self-harm, impulsivity, negative emotions, etc.) along with the core self-harming behaviors in the measuring tools. Second, the instruments on suicide attempts were not included in the current study. Since the goal of this study was to develop a measure for screening the self-harm behaviors and not suicide attempts, only the measures for self-harm and other related psychological variables, including depression, anxiety and self-esteem, were included. Considering the continuum of self-harm behaviors and suicide, additional information on suicide might extend the implications of our results.

In sum, the SHSI may be a useful screening measure for self-harm behaviors. Not only does it include the widely known core self-harming behaviors, but it is also culturally sensitive and comprises unique content. In this study, the SHSI was found to be a reliable and valid measure of NSSI in the Korean adolescent population. Nevertheless, this instrument just completed its initial development and validation, and additional research in the broader population is needed to examine its utility.

## Supporting information

**S1 Table. The Self-Harm Screening Inventory (SHSI).**
(XLSX)

**S2 Table. The Preliminary 28 items of the self-harm behaviors among adolescents.**
(XLSX)

## Author Contributions

**Conceptualization:** Sojung Kim, Dong Gi Seo, Jae-Chul Park, Yeonkyeong Son, Ji-Hyun Lee, Dasol Yoon, Jae-Won Kim, Jae Hyun Yoo, Jong-Sun Lee.

**Data curation:** Sojung Kim, Dong Gi Seo, Jong-Sun Lee.

**Formal analysis:** Dong Gi Seo, Ji-Hyun Lee.

**Funding acquisition:** Jong-Sun Lee.

**Investigation:** Jae-Chul Park, Yeonkyeong Son, Dasol Yoon, Jong-Sun Lee.

**Methodology:** Sojung Kim, Dong Gi Seo, Jae-Chul Park, Yeonkyeong Son, Jae-Won Kim, Jae Hyun Yoo, Jong-Sun Lee.

**Project administration:** Jong-Sun Lee.

**Supervision:** Jong-Sun Lee.

**Validation:** Sojung Kim, Dong Gi Seo, Jong-Sun Lee.

**Visualization:** Sojung Kim.

**Writing – original draft:** Sojung Kim, Ji-Hyun Lee.

**Writing – review & editing:** Sojung Kim, Dong Gi Seo, Jae-Chul Park, Yeonkyeong Son, Ji-Hyun Lee, Dasol Yoon, Jae-Won Kim, Jae Hyun Yoo, Jong-Sun Lee.

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
