## [Decision Letter · Decision Letter 0]

16 Sep 2021

PONE-D-21-17797Development and validation
of the Self-Harm Screening Inventory (SHSI)PLOS ONE

Dear Dr. Lee,

Thank you for submitting your manuscript to PLOS ONE. After careful consideration, we
feel that it has merit but does not fully meet PLOS ONE’s publication criteria as it
currently stands. Therefore, we invite you to submit a revised version of the
manuscript that addresses the points raised during the review process.

This Academic Editor recommend you to follow the recommendations provided by the
three Reviewers to improve your manuscript.

Please submit your revised manuscript by Oct 31 2021 11:59PM. If you will need more
time than this to complete your revisions, please reply to this message or contact
the journal office at plosone@plos.org. When
you're ready to submit your revision, log on to https://www.editorialmanager.com/pone/ and select the 'Submissions
Needing Revision' folder to locate your manuscript file.

Please include the following items when submitting your revised
manuscript:A rebuttal letter that responds to each point raised by the academic
editor and reviewer(s). You should upload this letter as a separate file
labeled 'Response to Reviewers'.A marked-up copy of your manuscript that highlights changes made to the
original version. You should upload this as a separate file labeled
'Revised Manuscript with Track Changes'.An unmarked version of your revised paper without tracked changes. You
should upload this as a separate file labeled 'Manuscript'.

If you would like to make changes to your financial disclosure, please include your
updated statement in your cover letter. Guidelines for resubmitting your figure
files are available below the reviewer comments at the end of this letter.

We look forward to receiving your revised manuscript.

Kind regards,

Prof. Paola Gremigni, Ph.D.

Academic Editor

PLOS ONE

Journal Requirements:

2. Please improve statistical reporting and refer to p-values as "p<.001" instead
of "p=.000". Our statistical reporting guidelines are available at https://journals.plos.org/plosone/s/submission-guidelines#loc-statistical-reporting.

3. Please provide additional details regarding participant consent. In the ethics
statement in the Methods and online submission information, please ensure that you
have specified what type you obtained (for instance, written or verbal, and if
verbal, how it was documented and witnessed). If your study included minors, state
whether you obtained consent from parents or guardians. If the need for consent or
parental consent was waived by the ethics committee, please include this
information.

5. We note that you have referenced (Walsh B, Frost A Unpublished study. 2005.which
has currently not yet been accepted for publication. Please remove this from your
References and amend this to state in the body of your manuscript: (Walsh B, Frost A
Unpublished study. 2005Unpublished study. 2005.as detailed online in our guide for
authors http://journals.plos.org/plosone/s/submission-guidelines#loc-reference-style 

Reviewers' comments:

Reviewer's Responses to Questions

**Comments to the Author**

1. Is the manuscript technically sound, and do the data support the conclusions?

Reviewer #1: Yes

Reviewer #2: Yes

Reviewer #3: Yes

2. Has the statistical analysis been performed
appropriately and rigorously? 

Reviewer #1: Yes

Reviewer #2: Yes

Reviewer #3: Yes

3. Have the authors made all data underlying the
findings in their manuscript fully available?

Reviewer #1: No

Reviewer #2: No

Reviewer #3: Yes

4. Is the manuscript presented in an intelligible
fashion and written in standard English?

Reviewer #1: Yes

Reviewer #2: Yes

Reviewer #3: Yes

5. Review Comments to the Author

Reviewer #1: I would like to express my sincere gratitude for the opportunity given
to review this manuscript on the development and the validation of the Self-Harm
Screening Inventory.

The study’s aim is clearly defined in terms of developing a tool that can help assess
or screen non-suicidal self-injury (NSSI) among adolescents in South Korea. The
study’s originality is unquestionable, as it varies from other scales or tools
developed to screen for NSSI. In addition, the study demonstrates adequate knowledge
and understanding of relevant studies on NSSI.

However, there are a few aspects of the manuscript that need clarifications.

In statements numbered 124 and 125 (pp 6), the authors state that existing measures
developed in Western countries have not been validated in Asian countries. This
seems to contradict the adaptation study by Kim et al. (2019) (statements 156
&157, pp7).

I stand to be corrected but to the best of my knowledge, 1st grade to 3rd grade
students are classified as elementary school pupils, not middle school (statements
136-138, pp 6)

In the procedure section, little information is given as to how students got to know
about the study. Please provide further details on that. In addition, the authors
state that monetary reward was given to the participants. Can the authors be
specific as to how much was given to each participant to be certain that
participants were not enticed to partake in the study but did volunteer.

At the reference section, there are a few “invalid citations” stated. Please review
the section and provide the appropriate reference.

Nonetheless, sentences are clearly expressed and readable.

Again, I appreciate the opportunity given to review this manuscript and hope to see
it in print.

Reviewer #2: The autor does an adequate job elucidating the methodology procedures
and data analysis in the manuscript. Overall, the paper is scientifically sound and
articulates the content well. However, there are a few questions / elements that
need to be addressed. Moreover, there are some mayor edits that are suggested to
enhance the text. Comments are provided below, in paragraphs.

The title is quite in line with the objective of the research paper; in this regard,
perhaps I would highlight the population since the validation wants to be sensitive
to it.

The abstract, lacks the hook and the literature gap that you are aiming to contribute
to through your paper. For example, it does not highlight the validation goal. It is
argued that there is dearth of psychological measures for non-suicidal self-injury
(NSSI) screening, but recent literature indicates that there are many and most are
for adolescents. In this respect I am not sure that SHI is the most widely used
instrument, it would need referencing. I suggest reviewing the current NSSI
instrument reviews. Recommend re-visiting it.

NSSI has been clearly defined, however it refers to self-injury as NSSI, self-harm,
and other terms. The literature has differentiated NSSI from deliberate self-harm
and suicide. If the objective is to screen for NSSI, the instrument should be
sensitive to non-suicidal self-injury, but it does not show to be so. I suggest
clarifying the terminology and how the instrument is suitable for NSSI
screening.

Overall, the introduction flows well and provides an overview on the remarkable
literature. However, I would consider revisiting the introduction and provide a
stronger principled argument explaining how your research question will be able to
address the gap. Especially regarding the existence of culturally sensitive items
and instruments have not been developed or validated in Asian countries.

Methods are generally clearly described and the instruments used seem mostly suitable
for their purpose. The statistical approach is adjusted to the development and
validation of psychological instruments. However, items have been eliminated in most
of the measures for this study. Different measures of NSSI and others have been
correlated. If items are removed from an already validated instrument, the data will
not be able to be compared between this study and others. Have they been removed
because the instrument was not validated for the age group or population of this
study? About validation, it would help to justify why it has been validated with the
variables with which it has been validated.

The discussion is according to the objectives and results. The selection criteria
based on expert has clearly delimited the items that are more serious, this could
make the SHSI measure less prevalence of NSSI than many of the commonly used
instruments. It would be necessary to demonstrate through literature the need to
agree on this criterion. On the other hand, it also seems necessary to make explicit
in the text that the instrument measures the most serious methods. Furthermore, it
should be discussed and also included in a more extended discussion of how SHSI
improves NSSI screening.

Finally, a few minor comments:

Line 285: “scree plot”

Line 314 and 320-322: The data is already in the table, perhaps it would help the
reader if it were evaluated instead of repeating the data.

Line 342: The FASM items selection process was described in in the author's doctoral
thesis.

Reviewer #3: This is very interesting paper regarding NSSI issues and wih this new
itmes, clinicinas can assess NSSI patients more systematically and effectively.
However, I have several concerns.

1. Please check reference 8 which is inapporpiriate. Also, the authors should correct
or replace the references regarding korean newspapers which is only written in
Korean and which is not found with using even internet adress such as ref. 14, 19,
20, and etc. The authors should support their ideas with more evidenced and
English-written publications, since this journal is internationally read.

2. In the introduction, I cannot understand what the authors mean by the difference
between self-injury and self-harm in page 3, line 64. Please clarify the meaning or
delete the sentence.

3. In the method section, I recommand the authors should show the readers how 28
items were selected from the scales the authors utilized, more clearly. For this
purpose, the authors can use the supplemental materials.

4. In the method section, the authors describled the exclusion and inclusion
processes they made. However, it is hard to read and understand full process
clerarly. With using addtional figure, the readers are able to understand your
process more clearly.

5. I also wonder what was first 20 items in detail.

6. PLOS authors have the option to publish the peer
review history of their article (what does this mean?). If published, this will
include your full peer review and any attached files.

If you choose “no”, your identity will remain anonymous but your review may still be
made public.

**Do you want your identity to be public for this peer review?** For
information about this choice, including consent withdrawal, please see our
Privacy Policy.

Reviewer #1: No

Reviewer #2: No

Reviewer #3: No

---

## [Author Response · Author response to Decision Letter 0]

8 Dec 2021

Response to the Reviewers

Dear Prof. Paola Gremigni,

I wish to resubmit an original article for publication in PLOS ONE, titled
“Development and validation of the Self-Harm Screening Inventory (SHSI) for
Adolescents.” The manuscript ID is PONE-D-21-17797.

The manuscript has been rechecked and the appropriate changes have been made in
accordance with the reviewers’ suggestions. The responses to their comments have
also been drafted and attached.

I thank the editor and the reviewers for their thoughtful suggestions and insights,
which have enriched the manuscript, and helped in producing a better and more
balanced account of the study. I hope that the revised manuscript is now suitable
for publication in your journal. 

I have also attached the certificate of English editing below. 

I thank you for your consideration, and am looking forward to hearing from you.

Sincerely,

Jong-Sun Lee 

Department of Psychology, Kangwon National University

Chuncheon-si 24341, Korea

jongsunlee@kangwon.ac.kr

Journal Requirements:

Response: We have checked the PLOS ONE’s style requirements and have used the
templates to format the manuscript’s title page and main body. 

2. Please improve statistical reporting and refer to p-values as "p<.001" instead
of "p=.000". Our statistical reporting guidelines are available at https://journals.plos.org/plosone/s/submission-guidelines#loc-statistical-reporting.

Response: As per your recommendation, we have improved the statistical reporting and
referred to p values as “p < .001”. 

3. Please provide additional details regarding participant consent. In the ethics
statement in the Methods and online submission information, please ensure that you
have specified what type you obtained (for instance, written or verbal, and if
verbal, how it was documented and witnessed). If your study included minors, state
whether you obtained consent from parents or guardians. If the need for consent or
parental consent was waived by the ethics committee, please include this
information.

Response: We thank you for your meticulous review. We have included the details
regarding the consent form and informed parental consent under the subsections on
“Participants” and “Procedures.” Please find the details below.

After: 

Participants (Page 6, Line 126-Page 7 Line 129)

We recruited participants on a nationwide scale from multiple elementary and middle
schools in South Korea. Participants were either introduced to this study during
their classes or were provided with the information on the bulletin board at school. 

Procedures (Page 10, Line 211-223)

All the participants voluntarily participated in the study. Participants and their
parents were informed about the purpose of the study and its procedures. The
research assistants provided additional information about the study on the request
of the participants or their parents. Written informed consent was obtained from the
participants and their parents prior to participation. A URL of the online survey
webpage was sent to the participants’ smartphones. Participants logged into a secure
webpage and completed a set of self-report measures containing the preliminary
versions of the SHSI, K-SHI, CDI, RCMAS, and RSES. It took them approximately 20
minutes to complete all the questionnaires, and they received a monetary reward
after submitting the measures. To estimate the temporal stability of the preliminary
SHSI, the same tool was administered to 97 participants, four weeks after the first
administration. Participants received a gift card worth 5,000 won (approximately 5
dollars) for their participation in the first administration and an additional 3,000
won (approximately 3 dollars) worth gift card after the second administration. 

Response: We have added the full statement in the Method section, including the full
name of the IRB and have provided information on the IRB approval, as stated
below.

Before

All research procedures were approved by the Institutional Review Board of the
authors’ university (IRB NO. KWNUIRB-2019-05-007-002).

After (Page 10, Line 223-Page 11, Line 225)

All research procedures were approved by the Institutional Review Board of the
Kangwon National University in Chuncheon, South Korea (IRB NO.
KWNUIRB-2019-05-007-002). 

5. We note that you have referenced (Walsh B, Frost A Unpublished study. 2005.which
has currently not yet been accepted for publication. Please remove this from your
References and amend this to state in the body of your manuscript: (Walsh B, Frost A
Unpublished study. 2005Unpublished study. 2005.as detailed online in our guide for
authors http://journals.plos.org/plosone/s/submission-guidelines#loc-reference-style 

Response: We appreciate your recommendation. We have removed the reference and
updated the body of our manuscript as well.

Discussion 

Before

Considering the purpose of a screening measure, the SHSI should reduce false-positive
errors in selecting adolescents with self-harming behaviors. The excluded four items
are not rare but less lethal and, interestingly, were not as popular as more serious
forms of self-harm (such as cutting with a knife, carving, hitting head, and
burning) in a clinical adolescent sample. (52) These results may be related to the
fact that our sample was recruited from a community population in early
adolescence.

After (Page 26, Line 475-478)

Considering that the previous findings suggest that serious self-harm methods are
used among adolescents in clinical setting [55], the SHSI—which considered factor
loading as the criteria and not the frequency—might have decreased the
false-positive errors in selecting the adolescents with self-harming behaviors. 

Reviewer #1: I would like to express my sincere gratitude for the opportunity given
to review this manuscript on the development and the validation of the Self-Harm
Screening Inventory.The study’s aim is clearly defined in terms of developing a tool
that can help assess or screen non-suicidal self-injury (NSSI) among adolescents in
South Korea. The study’s originality is unquestionable, as it varies from other
scales or tools developed to screen for NSSI. In addition, the study demonstrates
adequate knowledge and understanding of relevant studies on NSSI. However, there are
a few aspects of the manuscript that need clarifications.

1. In statements numbered 124 and 125 (pp 6), the authors state that existing
measures developed in Western countries have not been validated in Asian countries.
This seems to contradict the adaptation study by Kim et al. (2019) (statements 156
&157, pp7).

Response: Thank you for your comment. We meant that there is a lack of validated
measures that include culturally sensitive items for self-harm in Asia. To clarify
the meaning of the text, we have included additional information on self-harm
measures in Asian countries as depicted below.

Before (Page 5, Line 123-Page 6 Line 125)

Finally, existing measures were developed in Western countries, and culturally
sensitive items have not been developed or validated in Asian countries.

After (Page 6, Line 107 )

Finally, the existing measures have primarily been developed in the Western
countries, and culturally sensitive items have been seldom developed or validated
for use in the Asian countries. In Korea, researchers have translated and validated
several measures of self-harm [28-30]. However, no Korean measure has been developed
to assess the self-harm behaviors in the Korean population. Likewise, the Japanese
version of the DSHI had been validated [31]. However, originally developed Japanese
measures to assess self-harm behaviors in Japan, are hard to seek. Although Chinese
research has reported a few self-harm measures developed in China, the validation of
the measures is rare [32-34]. Considering the potential gap between self-harm
behaviors in the Asian and Western cultures, and the lack of reliable tools for
assessing specific self-harm behaviors in Asia, it seems necessary to develop items
reflecting self-harm behaviors in the Eastern countries.

2. I stand to be corrected but to the best of my knowledge, 1st grade to 3rd grade
students are classified as elementary school pupils, not middle school (statements
136-138, pp 6)

Response: In South Korea, the grade system is different from that of Western
countries. It consists of 1st–6th grade in elementary school, 1st–3rd grade in
middle school, and 1st–3rd grade in high school. Therefore, the 1st–3rd grade in
middle school in South Korea is comparable to 7th–9th grade in Western countries. We
have added the information in the manuscript to improve readers’ understanding as
below.

After (Page 7, Line 132-133)

Grade 1-3 in middle school in South Korea is comparable to the grade 7-9 in middle
school in Western countries. 

3. In the procedure section, little information is given as to how students got to
know about the study. Please provide further details on that. In addition, the
authors state that monetary reward was given to the participants. Can the authors be
specific as to how much was given to each participant to be certain that
participants were not enticed to partake in the study but did volunteer.

Response: We thank you for your invaluable advice and have improved on the Method
section by incorporating the suggested changes. In the Procedures section, we have
added details on advertisement and recruitment, as mentioned below. We have also
included the specific amounts for the monetary rewards received by the
participants

Participants

Before

We recruited participants on a nationwide scale in South Korea.

After (Page 6, Line 127-Page 7, Line 129)

We recruited participants on a nationwide scale from multiple elementary and middle
schools in South Korea. Participants were either introduced to this study during
their classes or were provided with the information on the bulletin board at school. 

Procedures

After (Page 10, Line 211-223)

All the participants voluntarily participated in the study. Participants and their
parents were informed about the purpose of the study and its procedures. The
research assistants provided additional information about the study on the request
of the participants or their parents. Written informed consent was obtained from the
participants and their parents prior to participation. A URL of the online survey
webpage was sent to the participants’ smartphones. Participants logged into a secure
webpage and completed a set of self-report measures containing the preliminary
versions of the SHSI, K-SHI, CDI, RCMAS, and RSES. It took them approximately 20
minutes to complete all the questionnaires, and they received a monetary reward
after submitting the measures. To estimate the temporal stability of the preliminary
SHSI, the same tool was administered to 97 participants, four weeks after the first
administration. Participants received a gift card worth 5,000 won (approximately 5
dollars) for their participation in the first administration and an additional 3,000
won (approximately 3 dollars) worth gift card after the second administration.

4. At the reference section, there are a few “invalid citations” stated. Please
review the section and provide the appropriate reference.

Response: We thank you for your comment. We have checked the invalid citations and
accordingly modified the manuscript as well. 

Nonetheless, sentences are clearly expressed and readable.Again, I appreciate the
opportunity given to review this manuscript and hope to see it in print.

Reviewer #2: The autor does an adequate job elucidating the methodology procedures
and data analysis in the manuscript. Overall, the paper is scientifically sound and
articulates the content well. However, there are a few questions / elements that
need to be addressed. Moreover, there are some mayor edits that are suggested to
enhance the text. Comments are provided below, in paragraphs.1. The title is quite
in line with the objective of the research paper; in this regard, perhaps I would
highlight the population since the validation wants to be sensitive to it.

Response: We thank you for your suggestion. We have included “for adolescents” in the
title, to highlight the population under study.

Before

Development and validation of the Self-Harm Screening Inventory (SHSI) 

After

Development and validation of the Self-Harm Screening Inventory (SHSI) for
Adolescents

2. The abstract, lacks the hook and the literature gap that you are aiming to
contribute to through your paper. For example, it does not highlight the validation
goal. It is argued that there is dearth of psychological measures for non-suicidal
self-injury (NSSI) screening, but recent literature indicates that there are many
and most are for adolescents. In this respect I am not sure that SHI is the most
widely used instrument, it would need referencing. I suggest reviewing the current
NSSI instrument reviews. Recommend re-visiting it.

Response: We thank you for your meticulous review. We have highlighted the validation
goal in the abstract, as mentioned below. Furthermore, we have reworded the text
highlighting the lack of culturally appropriate tools for assessing NSSI among Asian
adolescents. We also revisited recent literature on self-harm measures and deleted
the phrase ‘the most’ (for the use of SHI) from the manuscript. 

Abstract

Before 

Despite the rapidly increasing rate of non-suicidal self-injury (NSSI) among
adolescents, there is a dearth of psychological measures for NSSI screening. This
study aimed to develop the Self-Harm Screening Inventory (SHSI), a scale for
screening NSSI among adolescents.

After (Page 2, Line 7-Line 9)

Despite the rapidly increasing rate of non-suicidal self-injury (NSSI) among
adolescents, there is a dearth of culturally appropriate psychological measures
screening for NSSI among the adolescents in the Asian countries. This study aimed to
develop and validate the Self-Harm Screening Inventory (SHSI), a culturally
sensitive and suitable scale for screening adolescents for NSSI.

Page 5, line.

Before

Among the validated self-report measures, the Self-Harm Inventory (23) is the first
and the most widely used measure for assessing self-harming behaviors.

After (Page 5, Line 83-34)

One of the valid self-report measures is the Self-Harm Inventory (SHI) [23]; it is
the first and widely used measure for assessing self-harm behaviors. 

3. NSSI has been clearly defined, however it refers to self-injury as NSSI,
self-harm, and other terms. The literature has differentiated NSSI from deliberate
self-harm and suicide. If the objective is to screen for NSSI, the instrument should
be sensitive to non-suicidal self-injury, but it does not show to be so. I suggest
clarifying the terminology and how the instrument is suitable for NSSI
screening.

Response: We appreciate your thoughtful comment. As the SHSI was developed for
screening NSSI, which refers to self-injury without intent to die, instructions were
clearly given to the participants. We have added the instructions in the Method
section, as stated below. 

Before

The Self-Harm Screening Inventory (SHSI) is a brief self-report measure for assessing
self-harm behaviors among adolescents. The SHSI consists of 10 binary items (yes/no)
which ask about self-harm behaviors within the past year. Cronbach’s alpha of the
SHSI in this sample was .795. 

After (Page 7, Line 138-147)

The Self-Harm Screening Inventory (SHSI)

The Self-Harm Screening Inventory (SHSI) is a brief self-report measure for assessing
self-harm behaviors among adolescents. The SHSI consists of 10 binary items (yes/no)
which inquire about one’s engagement in self-harm behaviors within the past year.
For the purpose of assessing NSSI among adolescents, the definition of NSSI was
provided at the end of the instructions as, “Non-suicidal self-injury refers to
deliberate self-harm behavior without the intention to die.” The participants were
instructed to choose “yes” if they had engaged in self-harm at least once in their
lifetime, and to choose “no” if they had not. The Cronbach’s alpha of the SHSI for
this sample was .795. The final version of the SHSI is presented in Supplementary
Material S1. 

4. Overall, the introduction flows well and provides an overview on the remarkable
literature. However, I would consider revisiting the introduction and provide a
stronger principled argument explaining how your research question will be able to
address the gap. Especially regarding the existence of culturally sensitive items
and instruments have not been developed or validated in Asian countries.

Response: We thank you for your suggestion, and have reviewed sufficient literature
to enhance the contents related to the existing culturally sensitive items and
instruments, in the Asian countries. Please find the changes below. 

Before

Finally, existing measures were developed in western countries, and culturally
sensitive items have been developed or validated in Asian countries.

After (page 6, Line 107-118 )

Finally, the existing measures have primarily been developed in the Western
countries, and culturally sensitive items have been seldom developed or validated
for use in the Asian countries. In Korea, researchers have translated and validated
several measures of self-harm [28-30]. However, no Korean measure has been developed
to assess the self-harm behaviors in the Korean population. Likewise, the Japanese
version of the DSHI had been validated [31]. However, originally developed Japanese
measures to assess self-harm behaviors in Japan, are hard to seek. Although Chinese
research has reported a few self-harm measures developed in China, the validation of
the measures is rare [32-34]. Considering the potential gap between self-harm
behaviors in the Asian and Western cultures, and the lack of reliable tools for
assessing specific self-harm behaviors in Asia, it seems necessary to develop items
reflecting self-harm behaviors in the Eastern countries.

5. Methods are generally clearly described and the instruments used seem mostly
suitable for their purpose. The statistical approach is adjusted to the development
and validation of psychological instruments. However, items have been eliminated in
most of the measures for this study. Different measures of NSSI and others have been
correlated. If items are removed from an already validated instrument, the data will
not be able to be compared between this study and others. Have they been removed
because the instrument was not validated for the age group or population of this
study? About validation, it would help to justify why it has been validated with the
variables with which it has been validated.

Response: We thank you for your thoughtful comment. Please find below, the response
for both your questions. 

1) Have they been removed because the instrument was not validated for the age group
or population of this study?

Response: No, this study aimed to develop a brief screening measure for self-harming
behaviors among adolescents. Thus, we selected the most relevant and informative
items among the original items to screen the participants for NSSI.

2)  About validation, it would help to justify why it has been validated with the
variables with which it has been validated.

-----Messick (1989) recommended five criteria for validation, as under content,
construct, responses, criterion, and consequence evidence. To validate the scale,
the criterion evidence was used. It is "validated with the variables with which it
has been validated"

If the developed scale is correlated with the scale which it has been validated, the
developed scale can be validated. 

Reference

Messick, S. (1989). Validity. In R. Linn (Ed.). Educational Measurement (3rd edition)
(pp.13-100). Washington, DC: American Council on Education.

6. The discussion is according to the objectives and results. The selection criteria
based on expert has clearly delimited the items that are more serious, this could
make the SHSI measure less prevalence of NSSI than many of the commonly used
instruments. It would be necessary to demonstrate through literature the need to
agree on this criterion. On the other hand, it also seems necessary to make explicit
in the text that the instrument measures the most serious methods. Furthermore, it
should be discussed and also included in a more extended discussion of how SHSI
improves NSSI screening.

Response: We thank you for your thoughtful comment. For the development of a quick
screening tool, we adopted two methods: qualitative investigation and quantitative
investigation. In the qualitative investigation, we asked the experts in the field
of self-harm and suicide, to rate the appropriateness, sensitivity and prevalence of
each of the NSSI behaviors. Their ratings were not based on the severity or
lethality of the behavior. After selecting 20 items based on the expert’s ratings,
we used exploratory factor analysis to determine the final 10 items. In the
quantitative investigation, we used factor loadings and not the frequency of each
item, as the selecting criteria. To clarify the procedures and the meaning, we have
modified the manuscript as well, as stated below. 

Procedures (Page 9, Line 179-

For the development of the preliminary SHSI, multiple sources of information were
reviewed. Previous research findings (academic sources), news, and social media
(such as Twitter, Instagram, Facebook, and Tumblr), describing self-harm behaviors
and related clinical observations, were investigated. The comments of experts on
self-harm and suicide attempts (non-academic sources) were also considered. Further,
the items assessing self-harm behaviors in the pre-existing self-harm measures, such
as the SHI, DSHI, and FASM, were collected. The focus of the literature review was
to generate a list of specific and observable NSSI behaviors. Thereafter, the
overlapping items and dimensions in the list were either combined or reorganized. 

A total of 28 items were developed for the first version of the preliminary measure
(Supplemental Material S2) and sent to 11 Korean experts—four clinical
psychologists, five psychiatrists, and two school counselors —in the areas of
self-harm and suicide. The experts rated individual items for their appropriateness
(in assessing self-harm among adolescents), sensitivity (to distinguish the
adolescents engaging in serious self-harm), and prevalence (of the behavior among
Korean adolescents). They rated each item on a seven-point Likert scale (1 = very
unlikely; 7 = very likely) for appropriateness and sensitivity, and a three-point
Likert scale (1 = low prevalence; 3 = high prevalence) for prevalence. They also
provided additional comments on the utility of the items. Thereafter, the mean
rating scores for each item were calculated. First, items with scores lower than 5
for appropriateness and sensitivity, and scores lower than 2 for prevalence, were
identified. Of the 28 items, eight items below the aforementioned scores were
identified under all three dimensions, one item under two of the dimensions, and six
items under only one dimension. Items identified in more than two of the dimensions
were excluded. Second, the informative value of the excluded items was reconsidered
in the ensuing discussion among the authors. Of the nine items identified under more
than two dimensions, the item on “overdose” was considered a distinctive and
comparatively common method of self-harm. It was, thus, re-included among the
preliminary items. Consequently, eight items with low scores on more than two of the
dimensions (appropriateness, sensitivity, and prevalence) were discarded before
administration (e.g., items on Drank excessively, Engaged in unhealthy sexual
relationships (e.g., having multiple partners), Had unsafe sex, Starved to the point
of harming your body (for a few days at least), Took laxatives, diuretics, etc. to
the point of harming your body, Rubbed skin on rough surfaces,
Swallowed dangerous substances (bleach, hand sanitizer, detergent, etc.), Got a
tattoo). Twenty items with higher scores and utility were retained as preliminary
items for the SHSI and administered to the participants along with other self-report
measures. 

Results (Page 16, Line 305-309)

The final structure showed a good model fit: χ² (119) = 207.189, p < .001, RMSEA =
.038, CFI = .964, TLI = .959, WRMR = 1.154. Factor loadings for the total items
ranged between .555 and .985. Among the 17 items, 10 items (preliminary items 1, 2,
3, 4, 6, 7, 15, 17, 19, and 20), which showed high factor loadings (greater than
0.7), were examined by two authors for content validity, and were selected for the
final version of the SHSI. 

Discussion 

Before (Page 27, Line )

Considering the purpose of a screening measure, the SHSI should reduce false-positive
errors in selecting adolescents with self-harming behaviors. The excluded four items
are not rare but less lethal and, interestingly, were not as popular as more serious
forms of self-harm (such as cutting with a knife, carving, hitting head, and
burning) in a clinical adolescent sample. (52) These results may be related to the
fact that our sample was recruited from a community population in early
adolescence.

After (Page 26, Line 475-478)

Considering that the previous findings suggest that serious self-harm methods are
used among adolescents in clinical setting [55], the SHSI—which considered factor
loading as the criteria and not the frequency—might have decreased the
false-positive errors in selecting the adolescents with self-harming behaviors. 

7. Finally, a few minor comments:Line 285: “scree plot”Response: We are unsure as to
what you are suggesting for the “scree plot”. Please let us know what your
recommendation is. 

Line 314 and 320-322: The data is already in the table, perhaps it would help the
reader if it were evaluated instead of repeating the data.

Response: We thank you for your suggestion and have modified the manuscript
accordingly.

Line 314

Before

The item-total correlations for each item ranged between .310 and .670, and
Cronbach’s alpha was .795. 

After (Page 21, Line 352-353)

The item-total correlations for each item were acceptable, ranging between .310 and
.670. The Cronbach’s alpha was good (α = .795). 

Line 320-322

Before

The SHSI had high correlations with the SHI (r = .775, p < .01), and showed
moderate correlations with the CDI (r = .489, p < .01) and RCMAS (r = .433, p
< .01). The SHSI negatively correlated with the RSES (r = -.399, p < .01;
Table 7). 

After (Page 21, Line 359-363)

The SHSI had high correlations with the SHI (r = .775, p < .01), and moderate
correlations with the CDI (r = .489, p < .01) and RCMAS (r = .433, p < .01).
The SHSI was negatively correlated with the RSES (r = -.399, p < .01; Table 7).
The results indicate that SHSI is a reliable and valid measure for NSSI, and it is
distinct from the other psychological measures assessing depression, anxiety, and
self-esteem.

Line 342: The FASM items selection process was described in in the author's doctoral
thesis.

Response: We thank you for your comment. We have looked up the doctoral thesis of the
author (Lloyd, 1998) and found that the author “established a list of
self-mutilative behaviors based upon review of previous research (Ross & McKay,
1979; Walsh & Rosen, 1988), as well as the authors’ clinical experience.”
However, detailed information on the process of selecting the items was not
presented in the doctoral thesis either.

Reviewer #3: This is very interesting paper regarding NSSI issues and wih this new
itmes, clinicinas can assess NSSI patients more systematically and effectively.
However, I have several concerns.1. Please check reference 8 which is
inapporpiriate. 

Response: We had cited the estimates of prevalence among clinical inpatient
adolescents in the reference 8 (Hamza et al., 2008). The cited part was as below. 

“Estimates of prevalence suggest that among clinical inpatient samples, as many as
21% of adults (Briere & Gil, 1998) and 30 to 40% of adolescents engage in NSSI
(Darche, 1990, Jacobson et al., 2008). NSSI is not only a clinical health concern,
however, as recent estimates based on community samples indicate that as many as 13
to 29% of adolescents (Baetens et al., 2011, Brausch and Gutierrez, 2010, Heath et
al., 2007, Ross and Heath, 2002) and 4–6% adults engage in NSSI (Briere and Gil,
1998, Klonsky, 2011).”

Also, the authors should correct or replace the references regarding korean
newspapers which is only written in Korean and which is not found with using even
internet adress such as ref. 14, 19, 20, and etc. The authors should support their
ideas with more evidenced and English-written publications, since this journal is
internationally read.

Response: We thank you for your thoughtful comment. Since the SHSI was developed as a
culturally sensitive measure for the adolescent population, we have not only cited
references written in English but also those written in Korean. Due to the limited
academic evidences on the novel phenomena related to NSSI (e.g., engaged in
bloodletting) among adolescents in Korea, especially those different from Western
findings, we have included Korean newspapers. We have checked the online address of
the articles once again and modified the reference as below. 

Before

16. Lee JH, Choe, Y. Y., Jeon, J. Y. . "self-harm experience" in 70,000 middle school
and high school students 2018. November, 11th. [Available from: http://www.hani.co.kr/arti/society/
society_general/869668.html.

19. W. YJ. Adolescents cousulted with self-harm has been tripled. Yonhap News. 2019.
05. 29.

21. Edujin. Is your 'friend' safe, now? Nachimban. 2018. 11. 09.

22. Song HS. Warning for upsurge of self-harming adolescents. Shindonga. 2019. April
7th.

After

16. Lee JH, Choe YY, Jeon JY. "Self-harm experience" in 70,000 middle school and high
school students. The Hankyoreh. 2018 Nov 11 [Cited 2021 Oct 15]. Available from:
https://www.hani.co.kr/arti/society/society_general/869668.html

19. Yang, JW. Adolescents consulted with self-harm have been tripled. The Yonhap
News. 2019 May 29 [Cited 2021 Oct 15]. Available from: https://www.yna.co.kr/view/AKR20190529078500005.

21. Moon YH, Is your ‘friend’ safe, now?. The Edujin Nachimban. 2018 Nov 9 [Cited
2021 Oct 15]. Available from: http://www.edujin.co.kr/news/articleView.html?idxno=20216

22. Song HS. Warning for upsurge of self-harming adolescents. The Shindonga. 2019 Apr
7 [Cited 2021 Oct 15]. Available from: https://shindonga.donga.com/3/all/13/1686942/1.

2. In the introduction, I cannot understand what the authors mean by the difference
between self-injury and self-harm in page 3, line 64. Please clarify the meaning or
delete the sentence.

Response: Thank you for your valuable comment. We have removed “self-harm” and
clarified the meaning of the sentence as below. 

Before

Girls are known to show NSSI earlier than boys are at higher risk of developing
self-harm during adolescence. NSSI declines over time, but approximately 20% of
adolescents who start NSSI maintain the behavior for more than five years, and it
often develops into a chronic and malignant practice

After (Page 3, Line 38-42)

Girls are known to show NSSI earlier than boys and are at a higher risk of developing
self-harm behaviors during adolescence [5,9,10]. NSSI declines over the course of
its development from adolescence to early adulthood. However, approximately 20% of
adolescents maintain the behavior for more than five years, and it often develops
into a chronic and malignant practice that lasts until adulthood [4]. 

3. In the method section, I recommand the authors should show the readers how 28
items were selected from the scales the authors utilized, more clearly. For this
purpose, the authors can use the supplemental enlisting materials.

Response: We thank you for your thoughtful suggestion. We have added the preliminary
28 items under the Supplemental Material S2. 

Before

For the development of the preliminary SHSI, multiple sources of information were
reviewed. Previous research findings (academic sources), news and social media
contents describing self-harm behaviors, and clinical observations and comments of
experts in self-harm and suicidal attempts (non-academic sources) were investigated.
Further, items from pre-existing self-harm measures, such as SHI, DSHI, and FASM,
were collected. Overlapping items and areas were combined or reorganized. A total of
28 items were developed for the first version of the preliminary measure, and sent
to 11 Korean experts in self-harm and suicide: four clinical psychologists, five
psychiatrists, and two school counselors.

After (Page 9, Line 179-192)

For the development of the preliminary SHSI, multiple sources of information were
reviewed. Previous research findings (academic sources), news, and social media
(such as Twitter, Instagram, Facebook, and Tumblr), describing self-harm behaviors
and related clinical observations, were investigated. The comments of experts on
self-harm and suicide attempts (non-academic sources) were also considered. Further,
the items assessing self-harm behaviors in the pre-existing self-harm measures, such
as the SHI, DSHI, and FASM, were collected. The focus of the literature review was
to generate a list of specific and observable NSSI behaviors. Thereafter, the
overlapping items and dimensions in the list were either combined or reorganized. 

A total of 28 items were developed for the first version of the preliminary measure
(Supplemental Material S2) and sent to 11 Korean experts—four clinical
psychologists, five psychiatrists, and two school counselors —in the areas of
self-harm and suicide. Furthermore, to ease comprehension among the readers, we have
added the preliminary 28 items under the Supplemental Materials. 

[Supplemental Material_ S2 Table 2]

S2 Table 2. The Preliminary 28 items of the self-harm behaviors among adolescents

No.

Item

1

Overdosed on drugs. 

2

Cut my body with sharp objects. 

3

Hit my body (e.g. hitting my body, such as the head, hard with my hands). 

4

Banged my head against a wall, desk, etc.

5

Hit things hard with my fist.

6

Drank excessively.

7

Scratched my body. 

8

Engaged in unhealthy sexual relationships (e.g., having multiple partners).

9

Had unsafe sex.

10

Starved to the point of harming my body (for a few days at least).

11

Took laxatives, diuretics, etc. to the point of harming my body.

12

Cut or carved something onto skin using a knife. 

13

Pulled my hair out. 

14

Picked or pinched my wound. 

15

Burnt my skin with fire (using a cigarette, match, or other hot objects).

16

Stuck objects underneath my fingernails or into my skin.

17

Bit parts of my body (e.g., mouth and lips)

18

Scratched my skin until it left scars.

19

Picked or peeled off my skin.

20

Stabbed my body with sharp or pointed objects. 

21

Cut holes in my body (e.g. not ear piercings or body piercings that are meant to
enhance beauty, but to make holes to harm myself) 

22

Slit or cut my body with sharp objects 

23

Carved words or symbols onto my body.

24

Rubbed my skin on rough surfaces.

25

Swallowed dangerous substances (bleach, hand sanitizer, detergent, etc.).

26

Strangled my neck. 

27

Engaged in bloodletting (drained blood from my body). 

28

Got a tattoo.

4. In the method section, the authors describled the exclusion and inclusion
processes they made. However, it is hard to read and understand full process
clerarly. With using addtional figure, the readers are able to understand your
process more clearly.

Response: We thank you for your recommendation, and have added a figure which depicts
the research flow, as shown below.

Fig 1. Flow of the research

5. I also wonder what was first 20 items in detail.

Response: We thank you for your question. Please find preliminary 20 items in Table 1
of the manuscript.

to the Reviewers-final.docx
---

## [Editor Report · Decision Letter 1]

4 Jan 2022

Development and validation of the Self-Harm Screening Inventory (SHSI) for
Adolescents

PONE-D-21-17797R1

Dear Dr. Lee,

We’re pleased to inform you that your manuscript has been judged scientifically
suitable for publication and will be formally accepted for publication once it meets
all outstanding technical requirements.

Kind regards,

Paola Gremigni, Ph.D.

Academic Editor

PLOS ONE
---

## [Editor Report · Acceptance letter]

25 Jan 2022

PONE-D-21-17797R1 

Development and validation of the Self-Harm Screening Inventory (SHSI) for
Adolescents 

Dear Dr. Lee:

I'm pleased to inform you that your manuscript has been deemed suitable for
publication in PLOS ONE. Congratulations! Your manuscript is now with our production
department. 

Kind regards, 

on behalf of

Prof. Paola Gremigni 

Academic Editor

PLOS ONE